# Development of an Image Registration Technique for Fluvial Hyperspectral Imagery Using an Optical Flow Algorithm

**DOI:** 10.3390/s21072407

**Published:** 2021-03-31

**Authors:** Hojun You, Dongsu Kim

**Affiliations:** 1IIHR—Hydroscience and Engineering, University of Iowa, Iowa City, IA 52242, USA; hojun-you@uiowa.edu; 2Department of Civil and Environmental Engineering, Dankook University, Yongin-si 16890, Korea

**Keywords:** fluvial remote sensing, hyperspectral imagery, optical flow, image registration

## Abstract

Fluvial remote sensing has been used to monitor diverse riverine properties through processes such as river bathymetry and visual detection of suspended sediment, algal blooms, and bed materials more efficiently than laborious and expensive in-situ measurements. Red–green–blue (RGB) optical sensors have been widely used in traditional fluvial remote sensing. However, owing to their three confined bands, they rely on visual inspection for qualitative assessments and are limited to performing quantitative and accurate monitoring. Recent advances in hyperspectral imaging in the fluvial domain have enabled hyperspectral images to be geared with more than 150 spectral bands. Thus, various riverine properties can be quantitatively characterized using sensors in low-altitude unmanned aerial vehicles (UAVs) with a high spatial resolution. Many efforts are ongoing to take full advantage of hyperspectral band information in fluvial research. Although geo-referenced hyperspectral images can be acquired for satellites and manned airplanes, few attempts have been made using UAVs. This is mainly because the synthesis of line-scanned images on top of image registration using UAVs is more difficult owing to the highly sensitive and heavy image driven by dense spatial resolution. Therefore, in this study, we propose a practical technique for achieving high spatial accuracy in UAV-based fluvial hyperspectral imaging through efficient image registration using an optical flow algorithm. Template matching algorithms are the most common image registration technique in RGB-based remote sensing; however, they require many calculations and can be error-prone depending on the user, as decisions regarding various parameters are required. Furthermore, the spatial accuracy of this technique needs to be verified, as it has not been widely applied to hyperspectral imagery. The proposed technique resulted in an average reduction of spatial errors by 91.9%, compared to the case where the image registration technique was not applied, and by 78.7% compared to template matching.

## 1. Introduction

The need for information collection with respect to river environmental, hydraulic, and hydrological characteristics has enabled the development of fluvial remote sensing (FRS), which refers to remote sensing in fluvial rivers [1,2,3]. The use of remote sensing for targets consisting of water is technically more difficult than for other objects because of the unique characteristics of water. Consequently, the development of FRS is recent owing to its inherent difficulties. Approximately 200 papers related to aspects of FRS, such as hydrology and water environments, have been published, representing only ~15% of all papers in the remote sensing field [4,5,6,7]; this indicates that remote sensing is not a widely adopted method for collecting river environmental, hydrological, and hydraulic data [8]. Mertes [9] and Marcus and Fonstad [10] emphasized that FRS was the only method that could provide continuous data for all basins and rivers. In other words, FRS has been identified as the only method suited to both holistically identifying natural phenomena and conducting qualitative and quantitative river analyses. To conduct research using FRS, unmanned aerial vehicles (UAVs) are preferred as an FRS platform because they can operate at low altitudes and collect data using fast cycles [11,12,13].

Red–green–blue(RGB) optical sensors are commonly used in FRS, enabling easier acquisition of images and easy post-processing using advanced computer hardware and vision technology [9,10]. However, the application of remote sensing with optical imagery has been limited because optical images only contain red, green, and blue light information, whereas the amount of information in hyperspectral imaging is considerably higher, including ultraviolet and infrared data. In other words, a camera that contains a hyperspectral sensor can precisely identify land cover, vegetation, and water quality through hyperspectral images, including spectroscopic properties from light spectrum regions not visible to the unaided eye [14,15,16]. In addition, the identification of multiple properties with single measurement data is possible, because hyperspectral images are composed of data from several tens of times the wavelengths of general optical images. Hence, hyperspectral imaging, which can divide the reflectivity of sunlight into successive wavelengths when taking photographs, has been applied to FRS.

However, hyperspectral image acquisition using an UAV-based hyperspectral sensor has various limitations. For example, when the existing remote sensing technique, commonly applied at high altitudes, is applied to UAV-based hyperspectral imaging, the low-altitude UAV or sensor observation results are more sensitive to spatial global positioning system (GPS) errors. Because the physical size of the pixels that compose the images is proportional to the altitude, they also become smaller in low-altitude observations [17,18], thus a relatively large error rate can occur because of spatial errors. To address this problem, a high-performance gimbal (to prevent shaking during flight in an UAV) or a high-performance GPS (to provide higher spatial accuracy) have been considered. These options reduce the flight performance of the UAV owing to the weight of the additional equipment, and separate modules or systems to manage them during flight are not available yet.

As the acquisition of UAV-based hyperspectral imagery for FRS is still in its early developmental stage, several limitations should be addressed because of the nature of hyperspectral images, which have extensive data structures that are difficult to analyze. In particular, spatial accuracy becomes low when the existing image registration technique for remote sensing is applied to UAV-based hyperspectral imaging; thus, because FRS requires high spatial accuracy, existing image registration techniques cannot support the application of hyperspectral imagery to FRS.

Image registration is used to improve the spatial accuracy of remote sensing results; it is used to convert two or more images, photographed in different coordinate systems, to the same coordinate system through transformation. Image registration can provide relatively high spatial accuracy to images obtained in different locations or positions. In this technique, the corners of the base image, which have high spatial accuracy, are identified, followed by the positions where these corners match those of the target image.

Among others, the template matching method has been the most-used image registration algorithm [19,20]. Template matching requires a search window and an interrogation area definition as parameters. After the search window and interrogation area are provided, the cross-correlation coefficients of the base and target images are calculated by moving the position of the interrogation area in the search window. The calculated cross-correlation coefficient can be used to determine a correlation map that has the same size as the search window, and the position closest to one from the correlation map becomes the corner of the target image corresponding to the corner of the base image. Thus, a geometric relationship is established using corners for multiple base and target images, and image registration is finally completed through two-dimensional (2D) image transformation.

However, this conventional image registration technique is complicated for automatic analyzers because of the complex analysis process, and it also introduces uncertainty into the results, as the process relies to an extent on user experience, as applied to the search window and interrogation area parameter definition. In particular, UAV-based hyperspectral images include large data files and complex structures, which increases the image registration processing time. To overcome this limitation, the objective of this study was to develop an image registration method using optical flow, which is faster to calculate and easier to automate than template matching.

Although no published cases have focused on improving the spatial accuracy of UAV-based hyperspectral imagery, several studies have introduced image registration to improve spatial accuracy, including those by Yang et al. [21], Wang et al. [22], and Sedaghat et al. [23]. However, the application of this technique to actual measured data and the associated analysis remain difficult, because most methods involve complex procedures such as artificial feature detection [24,25], artificial intelligence, and machine learning. These methods have been increasingly applied to various remote sensing areas, as they are good alternatives for enhancing the traditional method of image registration. However, considering that low-altitude UAV-based hyperspectral images acquired from the fluvial domain have a very high spatial resolution (pixel size of approximately 7 cm) and thereby require a considerably heavy memory size (more than two-digits giga byte), a simple and physical method for hyperspectral images is required, preferably by modifying the conventional methods rather than using data-driven methods such as machine learning. Moreover, actual in-situ measurements, typically using RTK-GPS(Real-time Kinematic Global Positioning System) in riparian rivers, require considerably more laborious field work with higher cost than that performed on land; thus, they are still insufficient for training in machine learning to satisfy the required accuracy.

Cases involving UAV-based hyperspectral imagery have not been reported, and most related studies focus on remote sensing spatial accuracy challenges from the hardware perspective. Although FRS requires high spatial accuracy, a relatively large error rate will be obtained if the existing remote sensing techniques used at high altitudes are applied to UAV-based hyperspectral imagery. In particular, template matching, which is used in conventional remote sensing, needs to be improved, as the spatial accuracy provided by this method is generally unsatisfactory for FRS. Therefore, the objective of this study was to overcome the long calculation times and automation complexities of template matching. We aimed to resolve the problems of result variations caused by user bias, automation difficulties, and the increased uncertainty incurred when hyperspectral imaging is used to apply the image registration technique. To maintain result consistency and reduce uncertainty, we applied image displacement tracking technology, based on the optical flow algorithm, and Harris corner detection technology [26,27] for extracting corners. It has been reported that optical flow can reduce the uncertainty caused by user inputs and is less affected by rotation and scale than template matching. This has been achieved because the calculation speed of optical flow is faster than that of template matching and only requires an interrogation area parameter [20,28,29,30,31]. Using UAV-based RGB and hyperspectral image data from a river system, we focused on the following research objectives: (1) introduction of Harris corner detection, which defines singularity, optical flow algorithm calculation of the displacement between RGB and hyperspectral images from the detected singularity, and a 2D transformation method to warp hyperspectral images to RGB images based on the calculated displacement; (2) suggestion of hyperspectral imaging with high resolution and high spatial accuracy by applying the method presented in this study to RGB and hyperspectral images measured in a real river system; (3) validation of the method presented in this study compared with only geometric correction and template matching as conventional methods; (4) application of the image registered hyperspectral image for water body detection; (5) discussions and conclusion, including limitations of the proposed method.

## 2. Methodology

### 2.1. Review of Template Matching Method for Image Registration

Currently, image registration techniques can be largely classified into feature-based image registration based on common features such as feature points, outlines, and edges, matched between two images, and intensity-based image registration, using the similarity of the spatial distribution of pixel values in images [32]. In the present study, we aimed to apply optical flow to intensity-based image registration methods and perform comparisons and verification using template matching, which is the most widely used image registration algorithm.

Template matching is a technique in which the positions of similar images among input images are found using correlation, and the corresponding points between two images are extracted. As shown in Figure 1a, template matching requires a search window and interrogation area as user-defined parameters; after these parameters are established, the cross-correlation coefficient of the base and target images can be calculated by changing the position of the interrogation area in the search window, as shown in Figure 1b. The calculated cross-correlation coefficient can be used to produce a correlation map of the same size as the search window, and the position closest to 1 in the correlation map becomes the corner of the target image, which corresponds to the corner of the base image. The statistics used to compare the similarity between the two areas in the interrogation area are the difference, normalized difference, correlation, and normalized correlation. As the latter has generally been acknowledged as the most accurate and fastest method [33,34], we extracted the feature points for image registration using template matching with normalized correlation. The normalized correlation can be calculated using Equation (1), and Equation (2) is the corresponding form to calculate the normalized correlation between two images:(1)$$\frac{\sum_{i=1}^{n}\left(x_{i}-\bar{x}\right)\left(y_{i}-\bar{y}\right)}{\sqrt{\sum_{i=1}^{n}{\left(x_{i}-\bar{x}\right)}^{2}{\left(y_{i}-\bar{y}\right)}^{2}}}$$
(2)$$\frac{\sum_{i=0}^{W}\sum_{j=0}^{H}\left(\left(I_{1}\left(x^{\prime }+i,y^{\prime }+j\right)-\bar{I_{1}})\cdot (I_{2}\left(x^{\prime }+i,y^{\prime }+j\right)-\bar{I_{2}}\right)\right)}{\sqrt[{}]{\sum_{i=0}^{W}\sum_{j=0}^{H}{\left(I_{1}\left(x^{\prime }+i,y^{\prime }+j\right)-\bar{I_{1}}\right)}^{2}\cdot \sum_{i=0}^{W}\sum_{j=0}^{H}{\left(I_{2}\left(x^{\prime }+i,y^{\prime }+j\right)-\bar{I_{2}}\right)}^{2}}}$$

In Equations (1) and (2), n indicates the total number of pixels, I(x,y) represents the pixel value corresponding to the x and y coordinates, $x^{\prime }$ and $y^{\prime }$ denote the x and y coordinates of the position being searched, respectively, $\bar{I}$ is the average pixel value of the template, and W and H are the horizontal and vertical pixel sizes of the template, respectively.

This template matching method requires considerable time for its automated analysis steps because of the complexity and number of calculations, and also has an inherent uncertainty resulting from the subjectivity of the user inputs required to set the search window and interrogation area parameters. UAV-based hyperspectral images have large data files and complex structures, which increases the time required for the image registration process. Furthermore, template matching changes the size of the object, particularly when an image is photographed at an angle, and the spatial distribution expressed by the light intensity of the object varies according to the size difference between the two images, hindering their tracking. Although various techniques have been applied to overcome these limitations [35,36], they have been either individually applied to specific remote sensing fields, or have not been available to the public, which hinders their general adoption.

### 2.2. Corner Detection

In this section, the corner detection algorithm is described. This algorithm, used in the new technique, consists of three states (flat, edge, and corner) for corner detection, and a table for distinguishing the states [26,37]. For corner detection, when the interrogation area (W) for a random point in the image, $\left(x_{i},\;y_{i}\right)$, is moved by u and v, the light intensity area deviation at point i can be expressed using Equation (3):(3)$$E\left(u,v\right)=\overset{}{\underset{\left(x_{i},y_{i}\right)\in W}{\sum }}{\left[I\left(x_{i}+u,\;y_{i}+v\right)-I\left(x_{i},y_{i}\right)\right]}^{2}$$
where E denotes the deviation in W, I indicates the pixel value of the image, and x and y are the x and y coordinates, respectively.

In addition, when Taylor expansion is performed for the deviation, after moving a random point, $\left(x_{i},\;y_{i}\right)$, and W, by u and v, and omitting the polynomial term, Equation (4) is obtained:(4)$$I\left(x_{i}+u,\;y_{i}+v\right)\approx I\left(x_{i},y_{i}\right)+\left[\begin{array}{cc} \frac{\partial I}{\partial x}\left(x_{i},y_{i}\right) & \frac{\partial I}{\partial y}\left(x_{i},y_{i}\right) \end{array}\right]\left[\begin{array}{c} u\\ v \end{array}\right]$$

Therefore, Equation (4) can be expressed as shown in Equation (5):(5)$$E\left(u,v\right)\approx \overset{}{\underset{\left(x_{i},y_{i}\right)\in W}{\sum }}{\left[I\left(x_{i},y_{i}\right)+\left[\begin{array}{cc} \frac{\partial I}{\partial x}\left(x_{i},y_{i}\right) & \frac{\partial I}{\partial y}\left(x_{i},y_{i}\right) \end{array}\right]\left[\begin{array}{c} u\\ v \end{array}\right]-I\left(x_{i},y_{i}\right)\right]}^{2}$$

Equation (5) can be rewritten as Equation (6), which can be expressed as a matrix equation (Equation (7)):(6)$$E\left(u,v\right)\approx \overset{}{\underset{\left(x_{i},y_{i}\right)\in W}{\sum }}\left[{\left(\frac{\partial I}{\partial x}u\right)}^{2}+{\left(\frac{\partial I}{\partial y}v\right)}^{2}+2\frac{\partial I}{\partial x}\frac{\partial I}{\partial y}uv\right]$$
(7)$$E\left(u,v\right)\approx \left[\begin{array}{cc} u & v \end{array}\right]\left[\begin{array}{cc} \sum^{}{\left(\frac{\partial I}{\partial x}\right)}^{2} & \sum^{}\frac{\partial I}{\partial x}\frac{\partial I}{\partial y}\\ \sum^{}\frac{\partial I}{\partial x}\frac{\partial I}{\partial y} & \sum^{}{\left(\frac{\partial I}{\partial y}\right)}^{2} \end{array}\right]\left[\begin{array}{c} u\\ v \end{array}\right]$$

As shown in Equation (8), if the 2 × 2 matrix in Equation (7) is M, and the eigenvalues of matrix M are $\lambda_{1}$ and $\lambda_{2}$, then $\lambda_{1}$ and $\lambda_{2}$ represent the size of each axis after representing the shape of W for a random point $\left(x_{i},\;y_{i}\right)$ on new axes, through principal component analysis. As shown in Figure 2, if both eigenvalues $\lambda_{1}$ and $\lambda_{2}$ for matrix M are small, the point is determined to be flat, whereas if both eigenvalues are large, it can be determined to be a corner, and if one of the two eigenvalues is large, the point is determined to be an edge [26]. Thus, a random point,$\;\left(x_{i},\;y_{i}\right)$, determined to be a corner, is extracted as a corner, and is given as a position where the optical flow will be analyzed later.
(8)$$M=\left[\begin{array}{cc} \sum^{}{\left(\frac{\partial I}{\partial x}\right)}^{2} & \sum^{}\frac{\partial I}{\partial x}\frac{\partial I}{\partial y}\\ \sum^{}\frac{\partial I}{\partial x}\frac{\partial I}{\partial y} & \sum^{}{\left(\frac{\partial I}{\partial y}\right)}^{2} \end{array}\right]$$

### 2.3. Optical Flow

The optical flow algorithm used in this study is shown in Figure 3. The flow velocity was directly calculated by calculating the x, y, and t (time) gradient images of the base and target images for the given W to determine the change rate for a random pixel. The theory and process of the optical flow algorithm are described as follows [38,39].

Assuming that the brightness value of an object does not change, as the movement of each image is very small, the equation for calculating pixel flow velocity can be defined as Equation (9):(9)I(x,y,t)=I(x+dx,y+dy,t+dt)
where I represents the pixel value of the image; x and y are the coordinates for the x and y directions, respectively; t indicates time; and dx, dy, and dt represent the changes in the x and y directions, and in time, respectively.

When the Taylor expansion is performed for the right term of Equation (9), and the polynomial term is omitted, the following equation is obtained:(10)$$I\left(x+dx,y+dy,t+dt\right)=I\left(x,y,t\right)+\frac{\partial I}{\partial x}dx+\frac{\partial I}{\partial y}dy+\frac{\partial I}{\partial t}dt+\cdots$$

To simultaneously satisfy Equations (9) and (10), the sum of the derivatives of the right-side term of Equation (9) must be zero. Hence,
(11)$$\frac{\partial I}{\partial x}dx+\frac{\partial I}{\partial y}dy+\frac{\partial I}{\partial t}dt=0$$

Therefore, the equation for calculating the flow velocity of a specific pixel is defined as shown in Equation (12):(12)$$\frac{\partial I}{\partial x}\frac{dx}{dt}+\frac{\partial I}{\partial y}\frac{dy}{dt}+\frac{\partial I}{\partial t}\frac{dt}{dt}=0\to I_{x}u+I_{y}v=-I_{t}$$
where $I_{x}$ and $I_{y}$ are the differential matrices of the x directions, respectively, u and v indicate the pixel velocities in the x and y directions, respectively, and $I_{t}$ denotes the time differential matrix of the image.
(13)AV=B
where
$$A=\left[\begin{array}{cc} \begin{array}{c} I_{x}\left(q_{1}\right)\\ I_{x}\left(q_{2}\right) \end{array} & \begin{array}{c} I_{y}\left(q_{1}\right)\\ I_{y}\left(q_{2}\right) \end{array}\\ \begin{array}{c} \vdots \\ I_{x}\left(q_{n}\right) \end{array} & \begin{array}{c} \vdots \\ I_{y}\left(q_{n}\right) \end{array} \end{array}\right],\;V=\left[\begin{array}{c} u\\ v \end{array}\right],\;\mathrm{and}\;B=-\left[\begin{array}{c} \begin{array}{c} I_{t}\left(q_{1}\right)\\ I_{t}\left(q_{2}\right) \end{array}\\ \begin{array}{c} \vdots \\ I_{t}\left(q_{n}\right) \end{array} \end{array}\right]$$
where $q_{n}$ is the $n^{th}$ pixel, and n is the number of pixels in the image.

The derivatives for the x and y directions and time derivative can be expressed as shown in Equations (14)–(16), respectively.
(14)$$I_{x}\left(x,y\right)=0.5\left[I\left(x+1,\;y\right)+I\left(x+1,y+1\right)-I\left(x,y\right)-I\left(x,y+1\right)\right]$$
(15)$$I_{y}\left(x,y\right)=0.5\left[I\left(x,\;y+1\right)+I\left(x+1,y+1\right)-I\left(x,y\right)-I\left(x+1,y\right)\right]$$
(16)$$I_{t}\left(x,y\right)=0.25\left[I\left(x,\;y,t+1\right)+I\left(x+1,y,t+1\right)+I\left(x,y+1,t+1\right)+I\left(x+1,y+1,t+1\right)\right]-0.25\left[I\left(x,y,t\right)+I\left(x+1,y,t\right)+I\left(x,y+1,t\right)+I\left(x+1,y+1,t\right)\right]$$

The optical flow algorithm can derive results quickly because the size of W is the only parameter that must be set, and it involves a relatively small amount of computation. In other words, compared to template matching, the optical flow algorithm has lower uncertainty (by removing user bias), faster calculation speed, and enhanced automation possibilities because fewer parameters are used. The main difference between template matching and optical flow is that the result of the template matching is the displacement, whereas it is the pixel velocity between two images for the optical flow. Thus, a modified optical flow algorithm can be applied as an alternative to the traditional optical flow method, as shown in Figure 4. 

In this study, considering that the denominator of the flow velocity component in each direction was dt, if this was converted to one frame, the optical flow result could be converted to displacement between the base and target images. Similar to the aforementioned template matching, the optical flow algorithm can be applied to image registration, enabling a quick determination of the relation between a corner in the base image (RGB) and its corresponding location in the target image (hyperspectral). More than four results in the relation between corners stemming from optical flow, and the 2D transformation described in Section 2.4 can be established to finalize the image registration of the targeted hyperspectral image on top of the base image. Given the advantages of optical flow compared with template matching, such as computational time, this study aimed to propose an optical flow method alternative to template matching for image registration, which has been rare up until now.

Figure 5 summarizes the overall step-by-step procedure for applying optical flow using image displacement tracking technology and Harris corner detection technology for tracking corners. By using the procedure described, corners are extracted from the given base image, and then the base and target images are matched by specifying an interrogation area near the corners. In sequence, the displacement is calculated by applying the modified optical flow algorithm suggested in this new technique to the extracted base and target images. Finally, image registration can be performed by converting the actual coordinates of the pixels corresponding to the corner positions of the base and target images to 2D images.

#### 2.4. 2D Image Transformation

2D image transformation involves combining corners generated from image processing into one file, and then fixing them to the same coordinate by either transforming the image or allocating coordinates to the image, as shown in Figure 6. To transform different image coordinates of two corresponding points that have the same positions expressed in unique coordinates to the same coordinate system, a transformation matrix is used. This matrix is derived from the principle of 2D image transformation and allows all images to be transformed in due course. Generally used 2D image transformation methods include rigid, similarity, affine, and perspective transformations.

Rigid transformation involves transforming the position and direction only and maintaining the shape and size of the original image. Similarity transformation allows the position, direction, and size to be changed while maintaining the shape of the image. Affine transformation involves transforming an image while preserving the straight lines and distance ratios. Perspective transformation involves transforming images while preserving their straight lines, distance ratios, and parallelism. It has been found that perspective transformation, which preserves the parallelism of lines and distances, is appropriate for use with line scanning imagery, as the photographed space is generally projected onto the surface around the sensor.

Considering the preservation of position, direction, distance ratio, and parallelism, the relationship between two corresponding points using perspective transformation can be expressed as shown in Equation (17). Equation (17) is a general equation for transforming coordinates (x, y) to coordinates $\left(x^{\prime },\;y^{\prime }\right)$. Homogeneous coordinates represent a projected area, based on the projection origin, and express coordinates in vector form. Thus, (x, y) is generally expressed in vector form (x, y,1). In homogeneous coordinates, the coordinate (x, y) can be understood as a line consisting of numerous points. Therefore, homogeneous coordinates can consider the invariant scale issue, which is advantageous for 2D transformation, using corresponding points between images of different scales. Expression in the homogeneous coordinates (x, y, 1) rather conventional (x, y) coordinates is particularly necessary for images acquired from UAVs, because their perspective changes in terms of altitude, resulting in scale issues in the image. For homogeneous coordinates, Equation (17) can be expressed as Equation (18).
(17)$$\left[\begin{array}{c} x^{\prime }\\ y^{\prime } \end{array}\right]=\left[\begin{array}{cc} a & b\\ c & d \end{array}\right]\left[\begin{array}{c} x\\ y \end{array}\right]+\left[\begin{array}{c} e\\ f \end{array}\right]$$
(18)$$A^{\prime }=\left[\begin{array}{c} x^{\prime }\\ y^{\prime }\\ 1 \end{array}\right]=\left[\begin{array}{ccc} \begin{array}{c} h_{11}\\ h_{21}\\ h_{31} \end{array} & \begin{array}{c} h_{12}\\ h_{22}\\ h_{32} \end{array} & \begin{array}{c} h_{13}\\ h_{23}\\ h_{33} \end{array} \end{array}\right]\left[\begin{array}{c} x\\ y\\ 1 \end{array}\right]=HA$$

In Equation (18), when the matrix equation is solved, $h_{33}$ is 1; thus, there are eight unknown values in total. This indicates that at least four corresponding points between 2D images are necessary to establish an equation for determining the unknown values and the elements of the matrix equation. The 2D transformation process using perspective transformation consists of normalization, homography calculation, and denormalization.

Assuming that there are four corresponding corners, if the physical coordinates of the corners of the target image are $P_{1}\left(x_{1},y_{1}\right)$, $P_{2}\left(x_{2},y_{2}\right)$, $P_{3}\left(x_{3},y_{3}\right)$, and $P_{4}\left(x_{4},y_{4}\right)$, and the physical coordinates of the base image are $P_{1}^{\prime }\left(x_{1}^{\prime },y_{1}^{\prime }\right)$, $P_{2}^{\prime }\left(x_{2}^{\prime },y_{2}^{\prime }\right)$, $P_{3}^{\prime }\left(x_{3}^{\prime },y_{3}^{\prime }\right)$, and $P_{4}^{\prime }\left(x_{4}^{\prime },y_{4}^{\prime }\right)$, the matrix equation can be established as Equation (20), using Equation (19). Each corresponding point has known physical coordinates based on the coordinate system of each image. Thus, when the inverse matrix of the first matrix is multiplied by the left and right terms of Equation (20) to calculate the homography parameters, Equation (21) is obtained. Because the inverse matrix can be calculated for a square matrix with the same number of rows and columns, if there are more than four corresponding corners, it is advisable to apply the least-squares method by multiplying a pseudo-inverse matrix by the left and right terms.
(19)$$x^{\prime }=\frac{h_{11}x+h_{12}y+h_{13}}{h_{31}x+h_{32}y+1},y^{\prime }=\frac{h_{21}x+h_{22}y+h_{23}}{h_{31}x+h_{32}y+1}$$
(20)$$\left[\begin{array}{cc} \begin{array}{c} \begin{array}{c} x_{1}\\ 0 \end{array}\\ \begin{array}{c} \vdots \\ x_{4}\\ 0 \end{array} \end{array} & \begin{array}{c} \begin{array}{c} y_{1}\\ 0 \end{array}\\ \begin{array}{c} \vdots \\ y_{4}\\ 0 \end{array} \end{array} \end{array}\begin{array}{cc} \begin{array}{c} \begin{array}{c} 1\\ 0 \end{array}\\ \begin{array}{c} \vdots \\ 1\\ 0 \end{array} \end{array} & \begin{array}{c} \begin{array}{c} 0\\ x_{1} \end{array}\\ \begin{array}{c} \vdots \\ 0\\ x_{4} \end{array} \end{array} \end{array}\begin{array}{cc} \begin{array}{c} \begin{array}{c} 0\\ y_{1} \end{array}\\ \begin{array}{c} \vdots \\ 0\\ y_{4} \end{array} \end{array} & \begin{array}{c} \begin{array}{c} 0\\ 1 \end{array}\\ \begin{array}{c} \vdots \\ 0\\ 1 \end{array} \end{array} \end{array}\begin{array}{c} \begin{array}{c} -x_{1}x_{1}^{\prime }\\ -x_{1}y_{1}^{\prime } \end{array}\\ \begin{array}{c} \vdots \\ -x_{4}x_{4}^{\prime }\\ -x_{4}y_{4}^{\prime } \end{array} \end{array}\begin{array}{c} \begin{array}{c} -y_{1}x_{1}^{\prime }\\ -y_{1}y_{1}^{\prime } \end{array}\\ \begin{array}{c} \vdots \\ -y_{4}x_{4}^{\prime }\\ -y_{4}y_{4}^{\prime } \end{array} \end{array}\right]\left[\begin{array}{c} h_{11}\\ h_{12}\\ h_{13}\\ h_{21}\\ h_{22}\\ h_{23}\\ h_{31}\\ h_{32} \end{array}\right]=\left[\begin{array}{c} x_{1}^{\prime }\\ y_{1}^{\prime }\\ x_{2}^{\prime }\\ y_{2}^{\prime }\\ \vdots \\ x_{4}^{\prime }\\ y_{4}^{\prime } \end{array}\right]$$(21)$$\left[\begin{array}{c} h_{11}\\ h_{12}\\ h_{13}\\ h_{21}\\ h_{22}\\ h_{23}\\ h_{31}\\ h_{32} \end{array}\right]={\left[\begin{array}{cc} \begin{array}{c} \begin{array}{c} x_{1}\\ 0 \end{array}\\ \begin{array}{c} \vdots \\ x_{4}\\ 0 \end{array} \end{array} & \begin{array}{c} \begin{array}{c} y_{1}\\ 0 \end{array}\\ \begin{array}{c} \vdots \\ y_{4}\\ 0 \end{array} \end{array} \end{array}\begin{array}{cc} \begin{array}{c} \begin{array}{c} 1\\ 0 \end{array}\\ \begin{array}{c} \vdots \\ 1\\ 0 \end{array} \end{array} & \begin{array}{c} \begin{array}{c} 0\\ x_{1} \end{array}\\ \begin{array}{c} \vdots \\ 0\\ x_{4} \end{array} \end{array} \end{array}\begin{array}{c} \begin{array}{c} 0\\ y_{1} \end{array}\\ \begin{array}{c} \vdots \\ 0\\ y_{4} \end{array} \end{array}\begin{array}{c} \begin{array}{c} 0\\ 1 \end{array}\\ \begin{array}{c} \vdots \\ 0\\ 1 \end{array} \end{array}\begin{array}{c} \begin{array}{c} -x_{1}x_{1}^{\prime }\\ -x_{1}y_{1}^{\prime } \end{array}\\ \begin{array}{c} \vdots \\ -x_{4}x_{4}^{\prime }\\ -x_{4}y_{4}^{\prime } \end{array} \end{array}\begin{array}{c} \begin{array}{c} -y_{1}x_{1}^{\prime }\\ -y_{1}y_{1}^{\prime } \end{array}\\ \begin{array}{c} \vdots \\ -y_{4}x_{4}^{\prime }\\ -y_{4}y_{4}^{\prime } \end{array} \end{array}\right]}^{-1}\left[\begin{array}{c} x_{1}^{\prime }\\ y_{1}^{\prime }\\ x_{2}^{\prime }\\ y_{2}^{\prime }\\ \vdots \\ x_{4}^{\prime }\\ y_{4}^{\prime } \end{array}\right]$$

After the calculation of the homography, as shown in Equation (21), the coordinates can be transformed between corresponding points using Equation (18). The process of coordinate transformation through homography involves multiplying the homography by the physical coordinates of the corners corresponding to the target image. That is, the homography must be multiplied by the physical coordinates of the corners of the target image. In contrast, the physical coordinates of the target image corresponding to the corners of the base image can be determined by multiplying the physical coordinates of the base image by the inverse matrix of the homography. The existence of homography enables the transformation of physical coordinates between corresponding corners, and also between all points in the coordinate systems of the two images. Therefore, coordinate transformation of other images can be performed based on one of the two images, which can be moved to the same coordinate system, and error analysis can be performed by finding the differences between the coordinates and the ground truth, based on the coordinates of two transformed images.

## 3. Study Area and Application

### 3.1. Study Area

Satellite and aerial photographs showing the location of the Cheongmi Stream (Wonsam-myeon, Yongin-si, Gyeonggi-do, South Korea) and the data collection site are presented in Figure 7. To apply the image registration technique, which is a hyperspectral imagery post-processing step, we selected the Cheongmi Stream as the target stream (see Figure 7a). The Cheongmi Stream is the first tributary of the Namhan River and has a sandy soil bed. The study area was approximately 2 km upstream from the confluence of the Namhan River, and the bend selected for our work is shown in Figure 7b. On a bend, the flow velocity at the site was relatively fast, with water ripples occurring. The data collection passage (Figure 7c) was selected as an area that was both easy to access and suitable for UAV operations; the target site measured 170 m horizontally and 230 m vertically, around the target stream. UAV-based data collection occurred in the area shown in Figure 7c, whereas the image area used for analysis in this study, excluding edge and shooting errors, is depicted in Figure 7d.

### 3.2. Data Collection Methods and Results

The objective of this study was to evaluate the accuracy of the proposed image registration technique in a remote sensing application compared to that of the conventional image registration. Thus, both methods were applied to hyperspectral images, and the results were compared. The data required for this study consisted of RGB optical images captured with a general optical camera and hyperspectral images measured using a hyperspectral sensor. The RGB optical and hyperspectral images were collected on 23 July 2019. RGB optical images were captured for approximately 25 min, from 11:50 to 12:05, and hyperspectral images were taken for approximately 1 h, from 12:40 to 13:30. The temperature was approximately 27 °C, the humidity was 77%, and it was relatively sunny. The wind direction and speed were WSW and 1.5 m/s, respectively, indicating that the conditions were suitable for UAV operation. The study area was photographed from the air using a UAV mounted with an optical camera, as shown in Figure 8a, and a hyperspectral sensor (Corning^®^ microHSI™ 410 SHARK) was mounted on another UAV, as shown in Figure 8b. We used a DJI Mavic Pro UAV for optical image collection and a DJI Matrice 600 Pro for hyperspectral image collection. The hyperspectral sensor used in this study was a line scan-type sensor, with a viewing angle of 29.5°, a spatial resolution of 700 pixels per line, and the ability to photograph 300 lines per second. This sensor was able to collect band images for 150 wavelength bands, from 400 to 1000 nm, in approximately 4 nm intervals, including the visible light region. For spectral characteristics, the sensor provided normalized reflectance, with a 10-bit radiation resolution, using the non-uniformity correction processing technique, which normalizes the reflection luminance. The sensor could be easily mounted on the UAV and was advantageous for georeferencing, as it contained both GPS inertial measurement unit modules and provided latitude and longitude for each hyperspectral image pixel, considering the position and attitude.

The UAV flew using a DJI automatic flight function. The UAV flight paths used in the optical and hyperspectral images are shown in Figure 8c. As shown in this figure, the UAV flew along nine flight paths at ~18 m intervals to capture as much of the study area as possible, considering the optical camera and hyperspectral sensor viewing angles. As a result, we acquired 188 optical images and hyperspectral imagery of nine swathes. During the flights, we flew the UAV as close to the water surface as possible (approximately 30 m) to acquire high-resolution imagery.

### 3.3. Application Results

The UAVs flew nine times over the study area, photographing it as thoroughly as possible, considering the viewing angles of the optical camera and hyperspectral sensor. In total, 188 RGB optical images were collected using an optical camera, and structure from motion (SfM) was used to acquire precise orthogonal images, considering the topography. SfM is a method used to create three-dimensional (3D) point clouds, using the perspectives available from imagery taken from multiple viewpoints [40], being an extension of conventional aerial photogrammetry, which generates a 2D numerical or digital elevation model using images taken from satellites or aircraft. SfM technology can be used to extract the standard digital terrain model, recording surface altitudes across a study area, and also objects on the surface, in 3D.

In the past, topographic data were captured using images continuously acquired, as remote sensing was performed mainly using aircraft or satellites. Currently, improved topographic survey results can be obtained using a large number of randomly taken high-resolution images acquired using UAVs [40,41]. SfM is known to be the most efficient and cost-effective method [42], and we used the ‘Pix4DMapper’ SfM application in our study.

Using the topography created via SfM, high-precision, high-resolution orthogonal images were acquired, as shown in Figure 9, by conducting projection, registration, and a mosaic of the 188 captured images. Although there can be spatial errors due to GPS inaccuracy, SfM accuracy is known to be <1 cm, which is two or three times more than the ground sampling distance (GSD: the physical distance covered per pixel, based on altitude [43]). Therefore, the RGB images taken at a low altitude, as was the case in this study, had a very small GSD, and were not different from RGB orthogonal images, which have a spatial accuracy of <1 cm. We considered that the error analysis could be performed without difficulty, even if an orthogonal image was adopted as ground truth.

Because the hyperspectral sensor used in this study took photographs using the line scanning method, a single-swath hyperspectral image, as shown in Figure 10a, could be obtained when the hyperspectral image was taken using a single path. In our work, a single-swath hyperspectral image consisted of a hyperspectral cube with approximately 150 bands, together with the latitude and longitude images that recorded the latitude and longitude corresponding to each hyperspectral image pixel. In this study, hyperspectral imagery was collected in nine passes over the study area, and a hyperspectral image swath dataset from multiple paths was constructed, as shown in Figure 10b. The location of each pixel was determined from the latitude and longitude images included in the collected swath hyperspectral image dataset.

The georeferencing results, in which a location was allocated to each pixel, and a mosaic that combined multiple images into one image were created, as can be seen in Figure 10c. The hyperspectral images collected and georeferenced in this study consisted of 150 bands, which covered the visible light region of approximately 400–1000 nm in ~4 nm intervals, with a spatial resolution of approximately 0.02.

Figure 11 is fundamental to this work and presents some of the corners extracted through corner detection after a partial section has been enlarged to identify clear spatial characteristics. To apply the technique proposed in this study, the RGB optical image in Figure 11a was set as the base image, and the georeferenced hyperspectral image in Figure 11b was set as the target image. The physical displacement of the corners of the study and base areas were then derived by applying the optical flow algorithm developed in our study, and the resulting 2D image transformation of the target image using the derived displacements is presented in Figure 11c. As shown in Figure 11b, when simple georeferencing was performed, the same objects were generated more than once because of the inaccurate latitude and longitude information available for overlapping areas. However, when the proposed technique was applied, as shown in Figure 11c, we obtained a result that was almost the same as the RGB orthogonal image obtained via normal image registration.

## 4. Verification Methods and Results

### 4.1. Verification Method

To examine the accuracy of the technique proposed in this study, we compared basic georeferenced hyperspectral imagery (‘just-georeferenced’ in the following) against hyperspectral imagery to which image registration had also been applied. The purpose of this comparison was to quantify the spatial error that occurred when only georeferencing was applied to a hyperspectral image. For image registration, template matching and the optical flow technique proposed in this study were applied. For each algorithm, the template matching search window was set to 128 × 128 pixels, and the interrogation area was set to 64 × 64 pixels. The interrogation area of the optical flow was set to 64 × 64 pixels. The sizes of the template matching search window and interrogation area were determined by trial and error, while gradually increasing in size. To perform the two algorithms under the same conditions, the optical flow algorithm interrogation area was set to be the same as that for the template matching algorithm.

Regarding the spatial positions used for comparing the spatial accuracy of the two algorithms, we used the corners extracted from the RGB orthogonal image that was assumed to be true. In total, 128 corners were extracted, and their positions are shown in Figure 12a. Figure 12b shows an RGB orthogonal image expanded from the area marked in red, as well as the corners, and those indicated in the just-georeferenced image, shown in Figure 12c, are at the corresponding positions. Thus, the difference between the pixel coordinates at the positions of the corresponding corners and the actual coordinates marked the spatial accuracy of the georeferenced image.

Figure 12d,e presents the image registration results obtained using the displacement measured by applying the template matching and optical flow techniques, respectively. The spatial accuracy of each technique was calculated using the aforementioned method. We used the difference between the pixel coordinates and the actual coordinates for a quantitative comparison of spatial accuracy, using maximum, minimum, and root mean square error (RMSE) statistics. The RMSE was calculated using the distance between the position of the RGB orthogonal image and the position corresponding to the result of applying each technique, for each corner, as in Equation (22):(22)$$RMSE=\sqrt{\frac{1}{n}\overset{n}{\underset{i=1}{\sum }}\sqrt{{\left(X_{RGB}-X_{i}\right)}^{2}+{\left(Y_{RGB}-Y_{i}\right)}^{2}}}$$
where *n* indicates the number of corners, $X_{RGB}$ and $Y_{RGB}$ for the $i^{th}$ corner represent the X and Y coordinates in the RGB orthogonal image, respectively, and $X_{i}$ and $Y_{i}$ represent the X and Y coordinates, respectively, in the image to which the compared technique is applied.

### 4.2. Verification Results

The spatial accuracy values of the just-georeferenced image, based on the RGB orthogonal image, the image registration based on template matching, and the technique proposed in this study can be seen in Table 1. The just-georeferenced hyperspectral image showed spatial accuracies within 8.15–92.15 pixels, which corresponded to distances of 0.1621–1.843 m when converted to the actual coordinate system, using the image resolution of 0.02/pixel. Using the same process, the range of the spatial accuracy of the image registration technique based on template matching was 0.03–5.70 pixels and 0.001–0.154 m in pixel and physical coordinate systems, respectively. The spatial accuracy of the method proposed in this study ranged from 0.05–3.24 pixels and 0.001–0.065 m in the pixel and physical coordinate systems, respectively. Regarding their RMSEs, the spatial accuracy of the just-georeferenced image corresponded to 11.73 pixels, and the images to which template matching and optical flow were applied yielded spatial accuracies of 4.43 and 0.94 pixels, respectively. In the real coordinate system, these results corresponded to 0.235 m for the just-georeferenced image, 0.098 m for template matching, and 0.019 m for optical flow.

### 4.3. Discussion

In this study, we extracted the corners required for image registration from the base image and introduced an optical flow technique to estimate their positions in the target image. The RMSE for the distance of image registration to which optical flow was applied decreased by an average of 91.9% and 78.7%, compared to the cases when only georeferencing was applied and when template matching was applied, respectively. In terms of the pixel coordinate system, an average of approximately 12 pixel errors for the entire image occurred when only georeferencing was performed, whereas an average of approximately one pixel error occurred when image registration was performed using optical flow.

These errors are illustrated in Figure 13. Figure 13a represents the hyperspectral image to which the technique proposed in this study was applied, and it appears almost identical to the RGB orthogonal image in Figure 12. Figure 13b shows the minimum, maximum, and RMSE results achieved by georeferencing, template matching, and optical flow techniques. In Figure 13c, the minimum, maximum, and RMSE errors are represented as the radii of a circle, which facilitates their quantitative comparison.

The spatial accuracy was almost identical to that of the RGB orthogonal image, even when considering that the RGB orthogonal image used as the ground truth for each technique contained errors. The average RMSE of approximately 2 cm was considered too high because the resolution of the hyperspectral image used in this study was only 2 cm per pixel. However, because the surface characteristics of the object to be measured change only slightly over a 2-cm spatial change, the proposed technique could be considered appropriate for remote sensing using hyperspectral imagery.

The optical flow algorithm proposed in this study also requires fewer computations than template matching. This is because the number of optical flows is only proportional to the number of pixels comprising the interrogation area, whereas for template matching, the number of computations required is proportional to the product of the number of pixels comprising the search window and the number of pixels comprising the interrogation area. In other words, the optical flow technique proposed in this study can perform image registration faster than template matching, and thus it is a good candidate for future automation research applications using UAV-mounted image registration technology.

The technique proposed in this study can be applied to hyperspectral images used in the remote sensing field. The application of remote sensing to rivers (FRS) requires that the data captured for the river have a certain level of spatial accuracy; hence, the hyperspectral imagery used for river measurement data must be sufficiently accurate. Our results have shown that the method developed in this study can provide a basis for FRS; therefore, it is necessary to indicate the results related to the hyperspectral images that have sufficient spatial accuracy and can be achieved through a simple application.

An example of hyperspectral imagery measured using the proposed technique is shown in Figure 14. Figure 14a,b presents the spatial results calculated for the normalized difference vegetation index (NDVI), which is generally used to classify vegetation in remote sensing, and the normalized differential water index (NDWI), which is used to classify the existence or absence of water. Equations (23) and (24) were used to calculate the NDVI and NDWI, respectively. NDWI calculation requires shortwave infrared (SWIR) wavelength data as an input; however, the hyperspectral sensor used in this study did not have a SWIR element and was not designed for water bodies. Thus, water body data were extracted using a modified NDWI, in which the water content in water bodies was measured using green and near-infrared wavelengths, as shown in Equation (24) [44]. The equations used for NDVI and NDWI are as follows:(23)$$NDVI=\frac{NIR-Red}{NIR+Red}$$
(24)$$NDWI=\frac{Green-NIR}{Green+NIR}$$
where NIR indicates the light intensity of the wavelength corresponding to the near-infrared, and Red and Green denote the light intensity of the wavelength corresponding to red and green, respectively.

The closer the NDVI (NDWI) is to 1, the more likely the area analyzed is to have vegetation (water), whereas the closer it is to − 1, the more likely the area analyzed is to lack vegetation (water). Therefore, to extract vegetation and water body areas using NDVI and NDWI, the areas with values higher than zero can be extracted after calculating the NDVI and NDWI. A spatial representation of the calculated NDVI values is illustrated in Figure 14a, with an enlarged view presented in Figure 14c for more accurate observation. When the results in Figure 14a,c are compared with the RGB orthogonal image in Figure 12, it can be observed that the vegetation areas, as well as their spatial distribution, are clearly distinguishable. Similarly, the calculated NDWI values are spatially shown in Figure 14b,d presents an enlarged view. When the results in Figure 14c,d are compared with the RGB orthogonal image in Figure 12, it can be observed that the water bodies could be extracted and clearly distinguished through the NDWI. Thus, the water body areas in the river could be clearly extracted, as shown in Figure 14f, by distinguishing the areas where the NDWI was higher than zero, as shown in Figure 14e.

## 5. Conclusions

With the introduction of FRS, a technique for achieving hyperspectral imagery, a spatial accuracy sufficient to allow river remote sensing was required. The spatial accuracy of the developed technique was quantitatively and qualitatively compared with both the just-georeferenced method and the template matching method used in conventional remote sensing. For this comparison, a RGB orthogonal image with a relatively high spatial accuracy was used as ground truth.

It was found that the error embodied in the technique proposed in this study was 91.9% and 78.7% lower, on average, than those achieved using the just-georeferencing method and the template matching methodology, respectively. In terms of the pixel coordinate system, an average of approximately 12 and one pixel errors for the entire image occurred when just georeferencing and optical flow were used, respectively. Thus, it was found that the hyperspectral imaging required image registration, because only the used of georeferencing was insufficient, and also that the technique proposed in this study showed better results than the template matching method. We also found that the proposed technique was faster, as it involved fewer computation steps than the conventional technique.

However, this study had some limitations. First, it is difficult to generalize the spatial accuracy of the proposed method, as it was applied to a single study area. Second, improved template matching methods are available, and although they are difficult to apply owing to their complex processes, the proposed technique should be compared to them. Finally, the actual spatial accuracy was not verified, as the error analysis in our work was performed by adopting the RGB orthogonal image for ground truthing purposes. Considering these limitations, as hyperspectral imaging has just been introduced in FRS, considerable financial resources and time would be required to investigate multiple areas. Moreover, there are no examples demonstrating that improved template matching techniques have been released or applied to hyperspectral imagery.

Considering these issues, our future research will involve reviewing a more general application of the new technique by applying it to multiple study areas, and by trialing and comparing various other image registration techniques. The proposed technique is a basic FRS technology, in which we acquired high spatial accuracy hyperspectral imagery from a real river and extracted water body data from this imagery. The proposed technique could accelerate the application of hyperspectral imagery to FRS, and we believe it could be applied to various river measurement fields involving UAV-based hyperspectral imaging in the future, including depth measurement, calculation of suspended sediment loads, algal bloom detection, and substrate material classification.

## Figures and Tables

**Figure 1 sensors-21-02407-f001:**
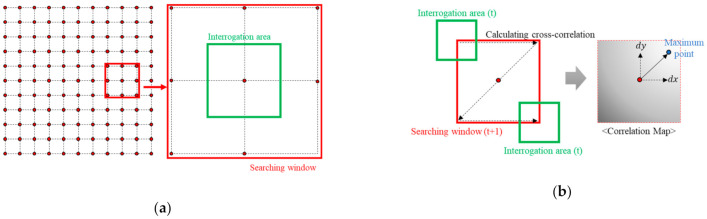
Conventional template matching technique: (**a**) configuration of the searching window and interrogation area, which are parameters of this technique; (**b**) calculation process of the correlation coefficient map using these parameters to find the displacement of the grid point in two images.

**Figure 2 sensors-21-02407-f002:**
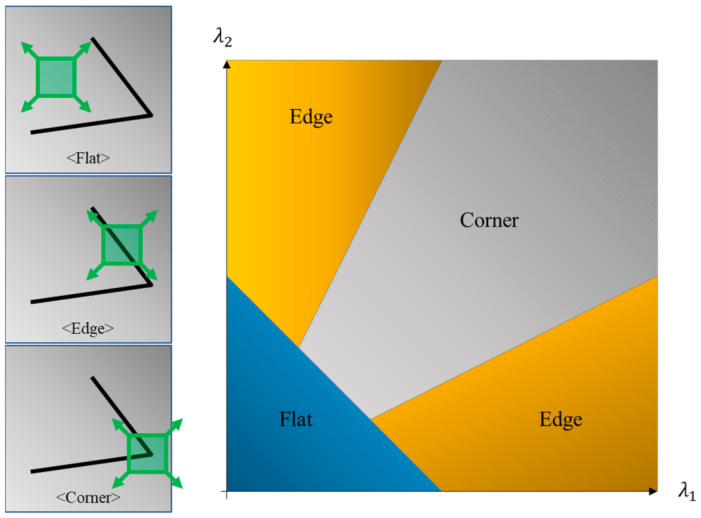
Corner type classification using the corner detection algorithm, and definition method for corner type using principal component of the matrix (M).

**Figure 3 sensors-21-02407-f003:**
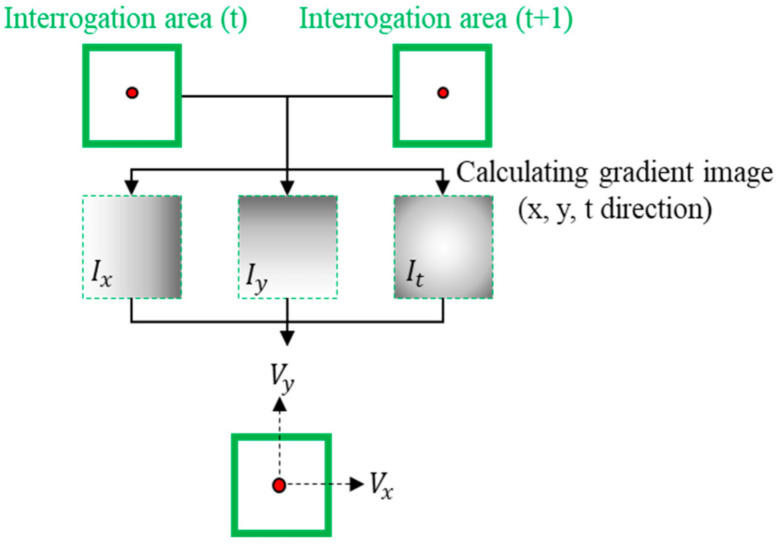
Procedure of the optical flow algorithm to calculate the velocity at a grid point.

**Figure 4 sensors-21-02407-f004:**
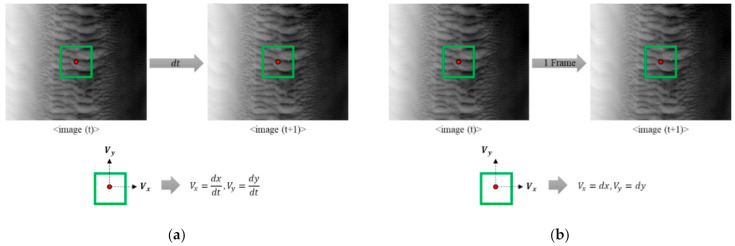
Comparison of the modified optical flow algorithm proposed in this study and the conventional optical flow algorithm: (**a**) modified and (**b**) conventional optical flow algorithm.

**Figure 5 sensors-21-02407-f005:**
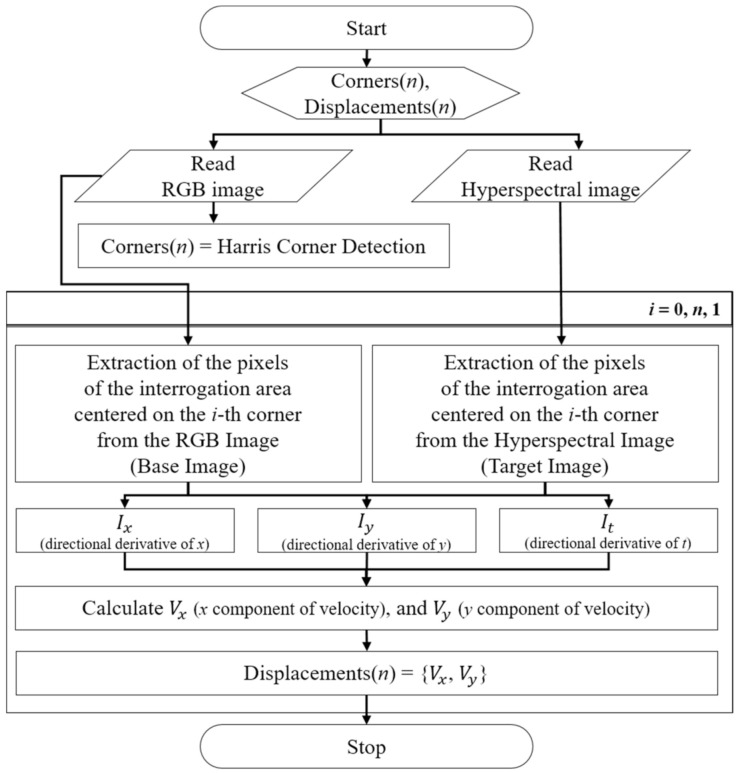
Process flow of the optical flow algorithm.

**Figure 6 sensors-21-02407-f006:**
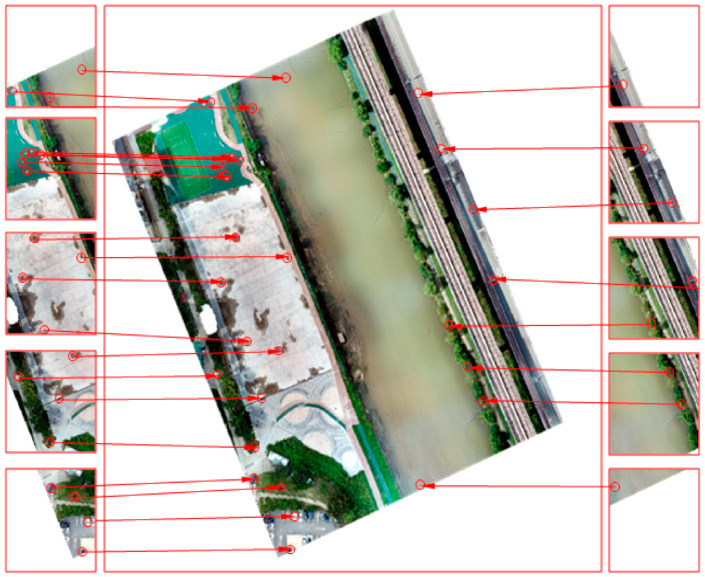
Schematic of 2D image transformation.

**Figure 7 sensors-21-02407-f007:**
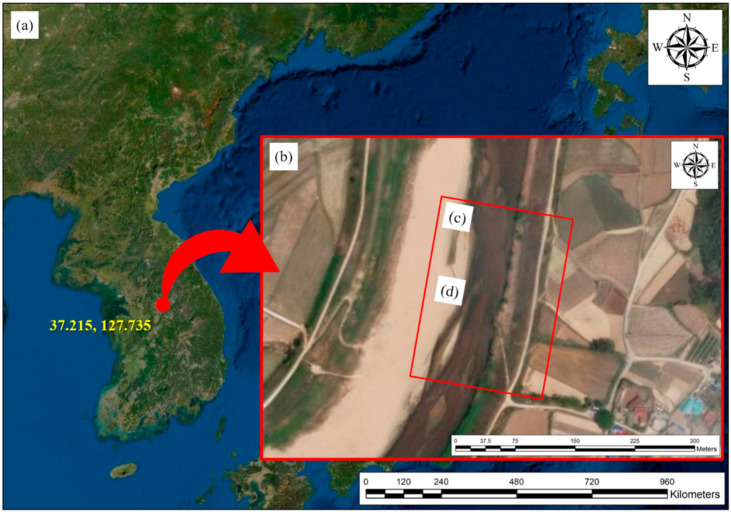
Study area, showing: (**a**) general location; (**b**) Cheongmi Stream site; (**c**) data collection passage; (**d**) image area used for analysis.

**Figure 8 sensors-21-02407-f008:**
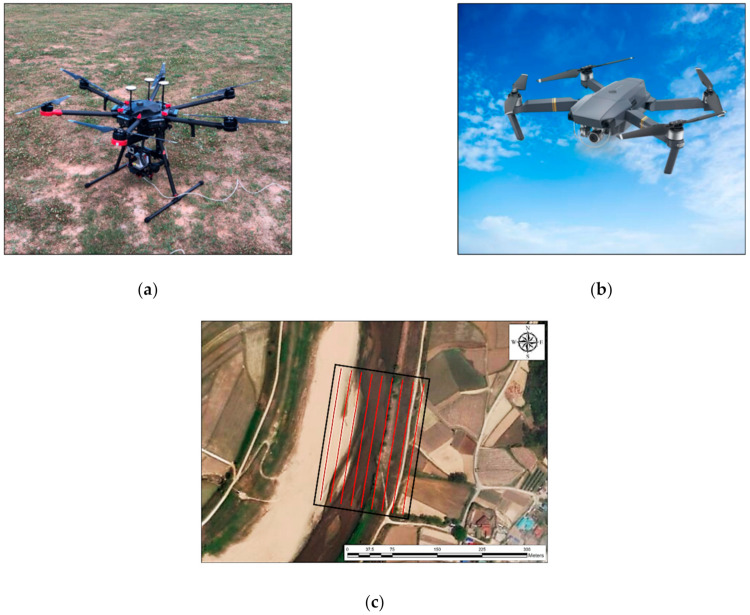
Data collection overview: (**a**) DJI Mavic Pro UAV, as used for optical image collection; (**b**) DJI Matrice 600 Pro UAV, as used for hyperspectral image collection; (**c**) red lines indicate the nine flight paths.

**Figure 9 sensors-21-02407-f009:**
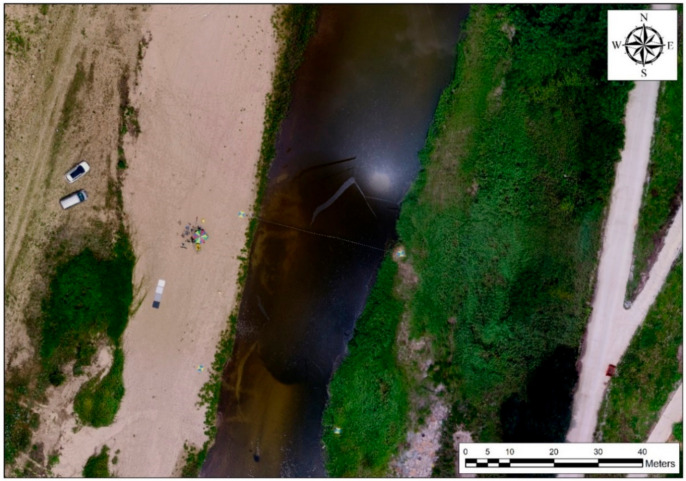
Example of use of the topography created via SfM (Structure from Motion) to capture high-precision, high-resolution orthogonal images.

**Figure 10 sensors-21-02407-f010:**
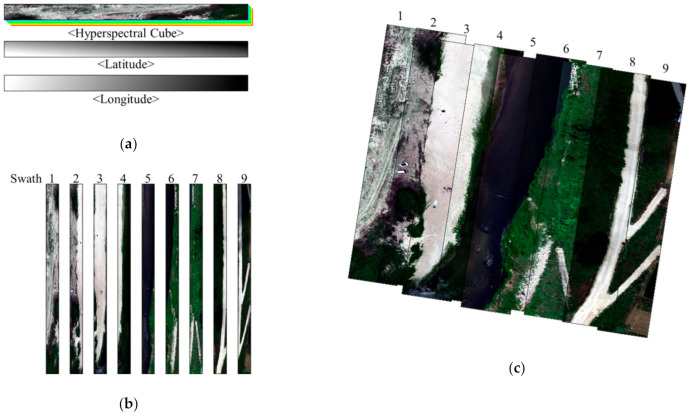
Overview of hyperspectral image data collection: (**a**) single-swath hyperspectral imagery; (**b**) hyperspectral image swath dataset created from multiple flights; (**c**) georeferencing results—in which a location was allocated to each pixel, and the mosaic that combined the multiple images into one image was created.

**Figure 11 sensors-21-02407-f011:**
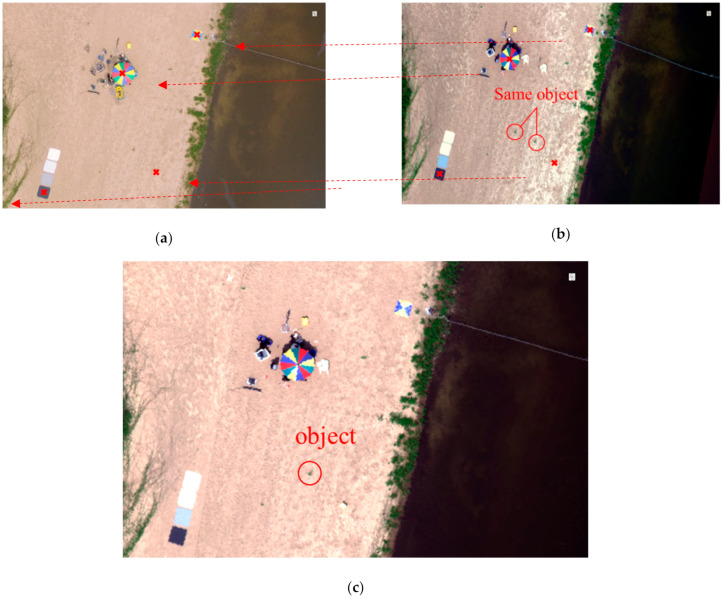
Example of application of the method proposed in this study: (**a**) RGB optical image; (**b**) georeferenced hyperspectral image; (**c**) 2D image transformation of the target image, achieved using the derived displacements.

**Figure 12 sensors-21-02407-f012:**
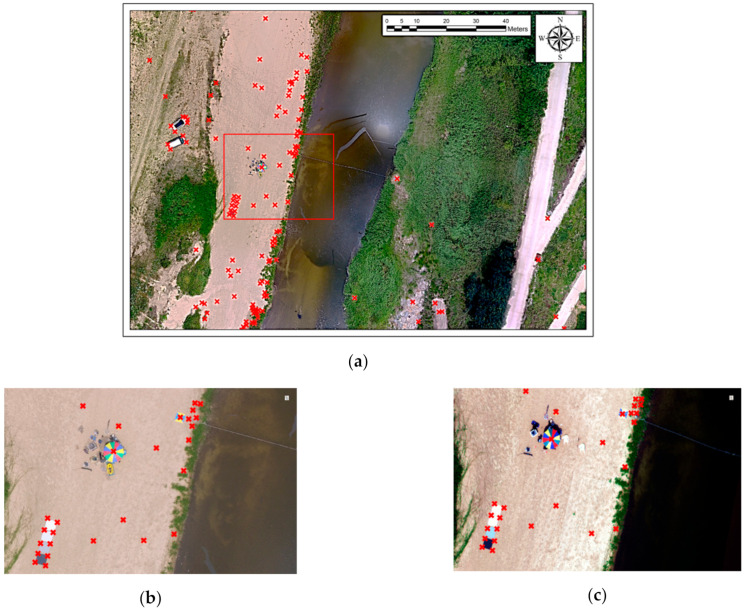
Datasets for verification: (**a**) 128 corners extracted from the RGB image used as ground truth; (**b**) RGB orthogonal image expanded from the area marked in red for qualitative comparison; (**c**) corners indicated in the image for which just georeferencing was applied; (**d**,**e**) image registration results achieved using the displacement measured by applying template matching and optical flow techniques, respectively.

**Figure 13 sensors-21-02407-f013:**
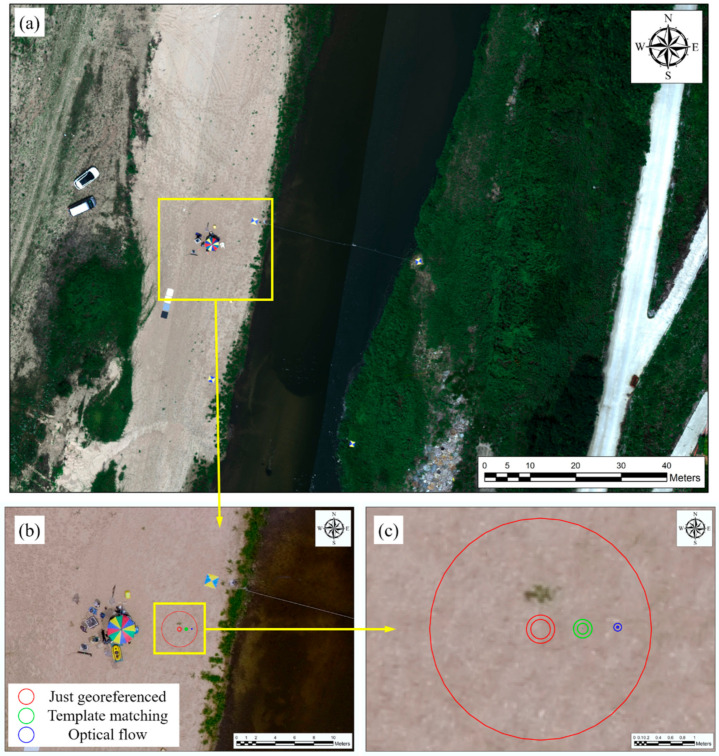
Comparison of spatial error size by each method: (**a**) hyperspectral image to which the technique proposed in this study was applied; (**b**) minimum, maximum, and RMSE results achieved by applying georeferencing, template matching, and optical flow; (**c**) minimum, maximum, and RMSE error comparisons for the (red) just-georeferenced, (green) template matching, and (blue) optical flow, using circle radius.

**Figure 14 sensors-21-02407-f014:**
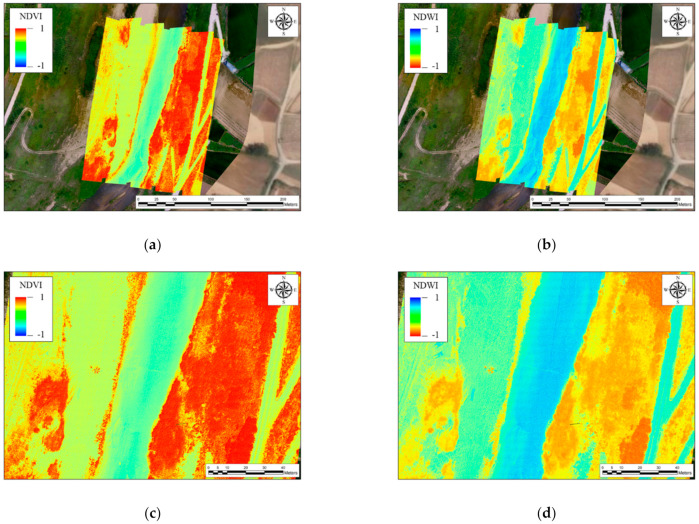
Hyperspectral imagery measured using the proposed technique: (**a**,**b**) spatial representations for results calculated for NDVI (Normalized Differential Vegetation Index) and NDWI (Normalized Differential Water Index), respectively; (**c**,**d**) respective enlargements; (**e**) area where the NDWI was higher than zero; (**f**) extracted river water body area.

**Table 1 sensors-21-02407-t001:** Spatial accuracy comparison between just-georeferenced imagery, based on the RGB orthogonal image, image registration based on template matching, and the technique proposed in this study.

Method (Elapsed Time)	Error					
	Minimum		Maximum		RMSE	
	Pixels	Actual (m)	Pixels	Actual (m)	Pixels	Actual (m)
Just-georeferenced	8.15	0.1621	92.15	1.843	11.73	0.235
Template matching (157.2 s)	0.03	0.001	5.70	0.154	4.43	0.089
Optical flow (19.5 s)	0.05	0.001	3.24	0.065	0.94	0.019

## Data Availability

The data that support the findings of this study are available from authors. Data are available from the author with the permissions.

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
