# Peer review of "Development of an Image Registration Technique for Fluvial Hyperspectral Imagery Using an Optical Flow Algorithm"

_sensors, 2021, doi:10.3390/s21072407_

Round 1

Author Response

This manuscript proposed an optical flow-based registration model of hyperspectral image. This topic is valuable in the field of remote sensing applications. The result is basically satisfactory. However, some other problems in the manuscript are still concerned in the following:

Q1. In the experiments, could the authors compare the proposed registration method with more state-of-the-art methods to validate the effectivity more extensively?

A1. In this manuscript, authors attempted to develop an efficient way to conduct image registration particularly against hyperspectral images with UAV-based high spatial resolution (less than 7 cm) acquired from the fluvial domine. We compared the proposed way with specifically respect to template matching which was very popular in the image registration used for conventional RGB images. We also tried to possibly maintain mathematically and physically based approach rather than relying upon black-box model such as machine learning and deep learning. So, these data-driven ways were not scope of the present study. Nevertheless, we do not deny the fact that these new state-of-the-art methods could be more promising that the proposed method. In this regard, we introduced such ways in the main text (line 187-194) rather than comparing them with our work.

Q2. How is the time cost of the proposed method compared to the related methods?

A2. We added the comparative result for the time cost among the proposed and related methods. Overall, the performance of the proposed methods was about 8 times better than template matching method.

Q3. Optical flow algorithm is very common for image registration. The authors should emphasize their contribution in Section 2

A3. As suggest, more emphasis about using optical flow algorithm for image registration of hyperspectral image was placed on section 2, which is as follows:

“Similarly with aforementioned template matching, optical flow algorithm allows its application to image registration where it enables to quickly find out a relation between a corner in base image (RGB) and its correspondent location in target image (hyperspectral). More than 4 resulted relation between corners stemmed from optical flow, 2D transformation described in the next section 2.4 can be established to finalize image registration of targeted hyperspectral image on top of the base image. Given the advantages of optical flow compared with template matching such as computational time, this study aimed to propose optical flow method alternatively than template matching for image registration, which has been rare up to now.” 

Q4. I suggest the authors to simplify the abstract.

A4. As suggested, the abstract becomes substantially simplified as much as we could. For example, we reorganized some part of introduction such as review for the template matching method to section 2. Besides, many unnecessary part like introduction of remote sensing was removed.

Q5. The organization of this manuscript should be added to the end of the introduction.

A5. As suggested, the organization of the present manuscript was added in the end of the Introduction.

“Using UAV-based RGB and hyperspectral image data from a river system, we pursue the research objectives with following sequence: 1) introduction of Harris corner detection that defines singularity and optical flow algorithm to calculate displacement between RGB and hyperspectral Image from the detected singularity, and 2-dimensional transformation method to warp hyperspectral image to RGB image based on calculated displacement; 2) suggestion of hyperspectral image with high resolution and high spatial accuracy by applying the method presented in this study to RGB and hyperspectral image measured in real river system; 3) validation of the method presented in this study comparing with only geometric correction and template matching as conventional method; 4) application of image registered hyperspectral image for waterbody detection; 5) discussions and conclusion including limitation of the proposed method.”

Q6. Please show the longitude and latitude of study area in Figure 7.

A6. Latitude and longitude were added in Figure 7.

Q7. Remote sensing image registration is very popular in the field. More works on remote sensing image registration should be included, such as “DOI: 10.1007/978-3-642-13681-8_13”, “DOI: 10.1016/j.isprsjprs.2019.03.002”…

A7. Authors appreciate the suggestion to include relevant literatures for image registration. It can be found on the line 117, it is each of the 24th and 25th references.

Reviewer 2 Report

This paper presents a hyperspectral image registration technique based on optical flow, and validate it on fluvial remote sensing images. The experimental results demonstrate that their methods have lower RMSE than just georeferenced and template matching. Overall, I think this is a good work for remote sensing in rivers. However, I have the following concern and suggestion.

  1. I have some concerns about the general usability of this work. The title of this paper is “Development of an image registration technique for Hyperspectral Imagery using an Optical flow algorithm.” However, it seems that you only validate this technique in fluvial remote sensing. It is unknown whether this method can be applied to other types of hyperspectral images. Therefore, I suggest narrowing down the title and focusing it on fluvial remote sensing or the authors can several datasets for validation.
  2. Machine learning and deep learning are two important methods in the field of image registration, but the relevant literature and introduction are too brief from lines 187 to 194. I suggest adding some contents here. Besides, why it is difficult to apply these techniques to the actual measured data? Please give some adequate explanations or evidences.
  3. In line 631, you claim that the optical flow method is faster than the template matching method. However, I did not find any comparisons of the speed specific to three methods. I suggest adding results of the calculation speed in Table. 1.
  4. The format and layout of this paper need to be refined. There are too may blank areas in page 6 and page 20. The sizes and positions of Fig. 10 (a), (b) and (c) also need to be adjusted.

Author Response

This paper presents a hyperspectral image registration technique based on optical flow, and validate it on fluvial remote sensing images. The experimental results demonstrate that their methods have lower RMSE than just georeferenced and template matching. Overall, I think this is a good work for remote sensing in rivers. However, I have the following concern and suggestion.

Q1. I have some concerns about the general usability of this work. The title of this paper is “Development of an image registration technique for Hyperspectral Imagery using an Optical flow algorithm.” However, it seems that you only validate this technique in fluvial remote sensing. It is unknown whether this method can be applied to other types of hyperspectral images. Therefore, I suggest narrowing down the title and focusing it on fluvial remote sensing or the authors can several datasets for validation.

A1. Authors appreciate the reviewer’s suggestion to narrow down the scope of the present method within fluvial domain. Obviously, authors should be careful to suggest Optical Flow method to hyperspectral images acquired from other domains. Therefore, we completely agree with this point suggested by the reviewer and changed the title accordingly by adding ‘fluvial’ to limit the application of the outcomes derived through this research.

Q2. Machine learning and deep learning are two important methods in the field of image registration, but the relevant literature and introduction are too brief from lines 187 to 194. I suggest adding some contents here. Besides, why it is difficult to apply these techniques to the actual measured data? Please give some adequate explanations or evidences.

A2. We more added relevant literature relevant to machine learning and AI techniques used for image registration. As the reviewer pointed out, we corrected and added more explanation regarding this issue. Therefore we addressed in the text that “it is difficult to apply these techniques to the actual measured data”. Nevertheless, authors do not deny the promise that machine learning or AI based way can show the better performance, although they are not currently scope of the present work. Authors admit that this part was a bit ambiguous, thus revised them as follows:.  

Page 4: “These methods are noticeable uprising approaches for the image registration increasingly applied for the various remote sensing area. It is clear these new ways are a good alternative enhancing the traditional way of image registration. However, considering that low altitude UAV-based hyperspectral images acquired from the fluvial domain revealed very high spatial resolution (around 7cm pixel size) and thereby requested considerably heavy memory size (more two-digit giga byte), simple and physically-based way to deal with such hyperspectral images is required and admitted in terms of modifying the conventional ways rather than data-driven way like machine learning. Moreover, actual in-situ measurements typically using RTK-GPS in the riparian rivers, which requires much higher laborious field works with expensive cost than land, are yet limited insufficiently enough for training machine learning to satisfy the accuracy.”

Q3. In line 631, you claim that the optical flow method is faster than the template matching method. However, I did not find any comparisons of the speed specific to three methods. I suggest adding results of the calculation speed in Table. 1.

A3. As suggested, we addition the calculation speed in Table 1. Overall, the comparative result demonstrated that the proposed method performed about 8 times faster than the conventional template matching method. 

Q4. The format and layout of this paper need to be refined. There are too may blank areas in page 6 and page 20. The sizes and positions of Fig. 10 (a), (b) and (c) also need to be adjusted.

A4. As the reviewer suggested, blanked areas were deleted after adjusting them.

Reviewer 3 Report

The manuscript deals with a method for registering images from different sensors (in the specific case RGB and hyperspectral taken from drone over river).

The manuscript is well written (even somewhat too verbose, starting from the very definition of remote sensing, but tolerable). The topic is interesting and results are convincing, therefore the manuscript can be considered for publication in the journal.

However I request that authors address some items, especially related to the maths in side the manuscript and to significance of their results. Actually Introduction creates too many expectations in the reader that are not satisfied indeed as authors clearly and honestly indicate some limitations of the study (see also considerations on the "proposed" Optical Flow method)

  • Authors have to revise Eq. (2), apparently wrong. From the formula terms with x' and y' go out from the sums and cancel at the numerator and denominator
  • Eq. (6) is wrong (u^2 and v^2 missing in the first two terms)
  • l. 332: I_t instead of It
  • Eqs. 14-15: it should be 0.5 instead of 0.25
  • l. 353-354. So, main contributions of the authors to Optical Flow is to set dt=1? If this is really the case, authors have to remove any claim about a proposed Optical Flow method. They can claim to have ported Optical Flow to the problem they face, but not to have proposed a variant of Optical Flow method
  • l. 389 and 400-401. Authors correctly affirm that at least 4 points are needed to solve the system (8 equations). If they use more points, then the system of equations is perfectly solvable with ordinary LS without using any pseudoinverse, that instead is mandatory in the case the system is undetermined (that is, one would use less than 4 points). In the case of overdetermined system, the pseudoinverse is equivalent to an ordinary inverse.
  • Eq. (18) expresses x', y' as a function of x, y, and already allows one to estimate coefficients h; it could be possible to reduce the number of equations to 2C instead of 3C (C is the number of corners). Authors prefer instead to revert to a form x, y as a function of x', y' (Eq. 20), that also allows one to estimate coefficients h. Is there any reason why authors revert to Eq. (20)?
  • Authors apply an algorithm initially to estimate corners in the images, both in the RGB and hyperspectral images, that are later projected through Optimal Flow algorithm. It is not clear how corners among RGB and hyperspectral images are matched. In other words, how a corner in an RGB image is associated to another corner in the hyperspectral image?
  • l. 726. Why Patents?
  • l. 676-678: The conclusion of the authors does not seem aligned with the work done. The fact the water is detected from the images through the NDWI indicator does not imply any consideration on the spatial resolution. 
  • l. 705: Authors honestly list some limitations of the study. It would be better to specify in the Introduction that this is a basic study, to somewhat reduce expectations of a reader.

Author Response

The manuscript deals with a method for registering images from different sensors (in the specific case RGB and hyperspectral taken from drone over river). The manuscript is well written (even somewhat too verbose, starting from the very definition of remote sensing, but tolerable). The topic is interesting and results are convincing, therefore the manuscript can be considered for publication in the journal.

However I request that authors address some items, especially related to the maths in side the manuscript and to significance of their results.

Q1. Actually Introduction creates too many expectations in the reader that are not satisfied indeed as authors clearly and honestly indicate some limitations of the study (see also considerations on the "proposed" Optical Flow method)

A1. It is true that Introduction highlighted the proposed optical flow method over its limitations. Also there were many redundance such that we substantially reduced and reorganized the introduction. Accordingly with the reviewer suggested, we also provide that the proposed optical flow method can be also limited:

“…; 5) discussions and conclusion including limitation of the proposed method. “

Q2-Q4. Authors have to revise Eq. (2), apparently wrong. From the formula terms with x' and y' go out from the sums and cancel at the numerator and denominator

Eq. (6) is wrong (u^2 and v^2 missing in the first two terms)

Line 332: I_t instead of It

Eqs. 14-15: it should be 0.5 instead of 0.25

A2~5. Authors appreciate the reviewer meticulously corrected our mistakes in the equation. We corrected them. Thank you.

Q5. l. 353-354. So, main contributions of the authors to Optical Flow is to set dt=1? If this is really the case, authors have to remove any claim about a proposed Optical Flow method. They can claim to have ported Optical Flow to the problem they face, but not to have proposed a variant of Optical Flow method

A5. The optical flow algorithm was firstly adapted to efficiently process image registration considering very heavy hyperspectral images rather substantially changing its algorithm. In this context, we reflected reviewer’s advice in Introduction and corrected them appeared in the manuscript to say that we did not change optical flow but applied it for image registration in order to save processing time and reduce input parameters for further automation. For instance, in section 2, we addressed that “we aimed to apply optical flow to intensity-based image registration methods,..”, where we used ‘apply’ rather than ‘develop’.  

Q6. l. 389 and 400-401. Authors correctly affirm that at least 4 points are needed to solve the system (8 equations). If they use more points, then the system of equations is perfectly solvable with ordinary LS without using any pseudoinverse, that instead is mandatory in the case the system is undetermined (that is, one would use less than 4 points). In the case of overdetermined system, the pseudoinverse is equivalent to an ordinary inverse.

A6. For the overdetermined system with more than 4 points, using LS is equivalent to use pseudoinverse. In that case, we use matrix for solving the LS, thereby needed pseudoinverse. Nothing more than this. Here the ordinary inverse is not useful because the matrix addressed in Eq.20 is not square matrix any more. Therefore, we can say that the pseudoinverse is not equivalent to an ordinary inverse in the case of overdetermined system. We revised this part in the manuscript to clarify its application. If less than 4 points, such case cannot be solvable definitely.

Q7. Eq. (18) expresses x', y' as a function of x, y, and already allows one to estimate coefficients h; it could be possible to reduce the number of equations to 2C instead of 3C (C is the number of corners).

A7. As described in line 377-387, Eq.18 was formulated to compensate the size issue appeared in the general coordinate for 2D transformation of Eq. 17. For the complete transformation correcting varying scale issue appeared in 3D domain (called homogeneous or perspective coordinate transformation), therefore, Eq.18 is needed as it is in conjunction with at least 4 points to build the relation in Eq. 18. In other words, through Eq.18, we can consider scale issue in perspective way. For instance, image acquired from UAV can be affected by its altitude, which results in scale issue definitely and they should be corrected in case that the altitude change during the drone monitoring. We revised the manuscript as follow to be clearer in this perspective.

“Therefore, homogeneous coordinates can consider invariant scale issue, making this characteristic advantageous for 2D transformation, using corresponding points between images of different scales. Expression in the homogeneous coordinates (x, y, 1) rather conventional (x, y) coordinate is particularly necessary for the images acquired from UAVs where their perspective in terms of altitude changes and results in scale issue in the image.”

Q8. Authors prefer instead to revert to a form x, y as a function of x', y' (Eq. 20), that also allows one to estimate coefficients h. Is there any reason why authors revert to Eq. (20)?

A8. Main purpose of describing Equation 19-21 was to demonstrate for the better understanding of users to more distinctively describe how conversion coefficient h can be solved based on the case with 4 referenced points. It should be noted that (x’, y’) and (x, y) indicate the coordinate of RGB image with geometrically corrected and hyperspectral images that we would like to transform, respectively. Eq.19 pointed out the relation for a reference point, and this relation becomes cumulated simultaneously for 4 points to result in Eq.20 as a matrix form. This is inevitable that Eq.20 can be formatted as the present form. For Eq.20, as well known, this procedure is conventional to resolve h (8 unknown) to make relation between (x, y) and (x’, y’) in Eq.19. We will eventually convert (x, y) in hyperspectral image accordingly with Eq.19 once h is figured out. 

Q9. Authors apply an algorithm initially to estimate corners in the images, both in the RGB and hyperspectral images, that are later projected through Optimal Flow algorithm. It is not clear how corners among RGB and hyperspectral images are matched. In other words, how a corner in an RGB image is associated to another corner in the hyperspectral image?

A9. Corners used as references points were searched with Harris method as depicted. But the corners were identified only for RGB image firstly, not for hyperspectral image. Then the optical flow algorithm was applied between the given corners from RGB to match them in the targeted hyperspectral image to find out where they are and how much they are departed. So, the corners become recognized in the hyperspectral image, which is the outcome of the given Optical Flow algorithm. This is a main idea of this article how the optical flow was applied for image registration. We reorganized Section 2 where Section 2.4 particularly summarized this aspect.

Q10. l. 726. Why Patents?

A10. This is definitely mistake. We cleared it out.

Q11. l. 676-678: The conclusion of the authors does not seem aligned with the work done. The fact the water is detected from the images through the NDWI indicator does not imply any consideration on the spatial resolution. 

A11. Yes, it is true that the waterbody detection through the NDWI is not outcome of main idea. And the spatial resolution is not relevant to the NDWI indicator. We corrected this part. Also we can point out that the NDWI indicator should be simply an example of how resulted hyperspectral image can be used especially for stressing the fact that the given domain of hyperspectral image used in this paper is a riparian area adjacent to rivers. We  removed this part to avoid confusion of readers as the reviewer pointed out. 

Q12. l. 705: Authors honestly list some limitations of the study. It would be better to specify in the Introduction that this is a basic study, to somewhat reduce expectations of a reader.

A12. Given limitations regarding proposed method, we revised the Introduction to reduce its impact in consideration of its limitation. Also, we specified and narrowed down its specified application to the fluvial domain and subsequently modified the title of the paper including ‘Fluvial’.

Page 4 “… 4) application of image registered hyperspectral image for waterbody detection; 5) discussions and conclusion including limitation of the proposed method.”

Round 2

Reviewer 1 Report

All my questions have been answered.

Reviewer 2 Report

The authors have addressed my comments.